# Complement Inhibitors for Geographic Atrophy in Age-Related Macular Degeneration—A Systematic Review

**DOI:** 10.3390/jpm14090990

**Published:** 2024-09-17

**Authors:** Ana Maria Dascalu, Catalin Cicerone Grigorescu, Dragos Serban, Corneliu Tudor, Cristina Alexandrescu, Daniela Stana, Sanda Jurja, Andreea Cristina Costea, Catalin Alius, Laura Carina Tribus, Dan Dumitrescu, Dan Bratu, Bogdan Mihai Cristea

**Affiliations:** 1Faculty of Medicine, “Carol Davila” University of Medicine and Pharmacy, 020021 Bucharest, Romania; ana.dascalu@umfcd.ro (A.M.D.);; 2Ophthalmology Department, Emergency University Hospital Bucharest, 050098 Bucharest, Romania; 3Faculty of Medicine, Titu Maiorescu University, 031593 Bucharest, Romania; 4Fourth Surgery Department, Emergency University Hospital Bucharest, 050098 Bucharest, Romania; 5Faculty of Medicine, Ovidius University, 900470 Constanta, Romania; 6Ophthalmology Department, Emergency County Hospital Constanta, 900591 Constanta, Romania; 7Department of Nephrology, Diaverum Clinic Constanta, 900612 Constanta, Romania; 8Faculty of Dental Medicine, “Carol Davila” University of Medicine and Pharmacy, 020021 Bucharest, Romania; 9Department of Internal Medicine, Ilfov Emergency Clinic Hospital, 022113 Bucharest, Romania; 10Faculty of Medicine, University “Lucian Blaga”, 550169 Sibiu, Romania; 11Department of Surgery, Emergency County Hospital Sibiu, 550245 Sibiu, Romania

**Keywords:** complement inhibitors, personalized therapy, age-related macular degeneration, geographic atrophy, biomarkers, outcomes, pegcetacoplan, lampalizumab, avacincaptad pegol

## Abstract

Background/Objectives: Age-related macular degeneration (AMD) is one of the main causes of blindness and visual impairment worldwide. Intravitreal complement inhibitors are an emergent approach in the treatment of AMD, which have had encouraging results. This systematic review analyzes the outcomes and safety of complement inhibitor therapies for GA in AMD cases. Methods: A comprehensive search on the PubMed and Web of Science databases returned 18 studies involving various complement inhibitor agents, with a total of 4272 patients and a mean follow-up of 68.2 ± 20.4 weeks. Results: Most treated patients were white (96.8%) and female (55.8%), with a mean age of 78.3 ± 7.8 years and a mean GA area of 8.0 ± 3.9 mm^2^. There were no differences in visual function change between treated and control participants. The mean GA area change was 2.4 ± 0.7 mm^2^ in treated participants vs. 2.7 ± 0.8 mm^2^ in control groups (*p* < 0.001). The ocular and systemic side effects were similar to those of intravitreal anti-VEGF. A less-understood effect was that of the onset of choroidal neovascularization (CNV) in 1.1–13% of patients; this effect was found to be more frequent in patients with neovascular AMD in the fellow eye or nonexudative CNV in the study eye at baseline. Conclusions: Complement inhibitors may represent a useful therapy for GA in AMD, but a personalized approach to patient selection is necessary to optimize the outcomes.

## 1. Introduction

Age-related macular degeneration (AMD) is one of the major causes of vision loss worldwide, with high-income countries experiencing the highest number of cases of AMD-related blindness [1,2]. Due to population, the incidence and prevalence of AMD continue to increase. A recent study by Wong et al. [1] found an estimated number of cases of any type of age-related macular degeneration of 196 million in 2020, rising to 288 million in 2040 [1]. Out of these, 1.85 million people are currently blind due to AMD, and another 6.23 million experience medium–severe visual impairment [2].

Previous research has found that AMD is a multifactorial disease, with genetic, metabolic, circulatory, and inflammatory factors being involved. The progressive degeneration and atrophy of photoreceptors, retinal pigment epithelium (RPE), and choriocapillaris lead to areas of geographic atrophy that significantly impair reading and driving and affect quality of life [3,4].

Intravitreal anti-VEGF therapy, the current gold standard in wet AMD, achieves disease stabilization and AV improvement in most cases. Dry AMD management is currently based on regular follow-up for early detection of choroidal neovascularization (CNV), addressing the risk factors, and prescribing supplements with dose antioxidant vitamins, lutein, zeaxanthin, copper, and zinc, based on the outcomes of the AREDS2 study in preventing progression [5,6,7]. However, the effect of nutritional supplements on AMD is still a subject of controversy. A recent systematic review and meta-analysis conducted by Evans et al. [5] found that supplements containing lutein and zeaxanthin may have little or no effect on the progression of AMD. However, in a comprehensive review, Pameijer et al. [6] supported the idea that a high intake of specific nutrients, the use of antioxidant supplements, and adherence to a Mediterranean diet were not associated with disease progression. The lack of effective therapy for dry AMD and its progression to geographic atrophy prompts the creation of new treatment modalities.

Several experimental studies proved the role of complement activation in damage and inflammation of the RPE. Moreover, clinical studies found higher levels of C3 and C5 fragments in the serum and vitreous of patients with AMD compared to controls [8,9,10]. Multiple complement fragments, such as C3a, C4a, C5a, complement factor P, CFH, and membrane attack complex (MAC), were located at the level of the drusen and the aqueous humor in patients with AMD [11,12]. Several experimental studies found that increased serum and vitreous complement proteins, as well as decreased regulatory complement factors, correlate with AMD. Genome-wide association studies (GWAS) have identified multiple loci and independently linked genetic variants for complement factor H (CFH), C2, complement factor B, C3, C9, and complement factor I that were associated strongly with an increased risk of AMD [13,14,15]. Klein et al. [15] identified a genetic variant of complement factor H (CFH), a natural inhibitor of complement activation, associated with a polymorphism located in a region of CFH that binds to both heparin and C-reactive protein (CRP) and decreases its binding efficiency [Klein]. Two common variants within CFH, Y402H and rs1410996 SNPs, were associated with a decreased affinity of binding complement fractions by CFH and a significant risk of AMD in the general population [16,17]. More than 90 genetic variants for C3 genes were documented to increase the risk of AMD. Among the most commonly encountered, R102G and K155Q variants were associated with resistance to proteolysis C3b by CFH and CFI, inducing further activation of alternate pathway activation by persistently increasing the concentrations of C3b [18,19].

Initially used in preventing hemolysis in paroxysmal nocturnal hemoglobinuria and lacking complement inhibitors in the membranes of the erythrocytes, complement inhibitors offer a promising approach for controlling inflammatory, immune, and degenerative diseases, such as an active antineutrophil cytoplasmic antibody (ANCA)-associated vasculitis, cold agglutinin disease, Neuromyelitis optica spectrum disorder (NMOSD) in adults who are anti-aquaporin-4 (AQP4) antibody positive, CD55-deficient protein-losing enteropathy (PLE), also known as CHAPLE disease, and recently geographic atrophy (GA) in AMD [20,21,22].

With two agents already having received FDA approval—Pegcetacoplan (Syfovre™) and Avacincaptad Pegol (IZERVAY™)—in 2023, and others under investigation in phase 2/3 trials, complement inhibitors seem to be a promising tool in the management of this debilitating disease [23]. The present paper is a systematic review that analyzes the indications, safety, and outcomes of complement inhibitors for GA in AMD.

## 2. Materials and Methods

A comprehensive search was carried out on PubMed and Web of Science using the mesh terms (“age-related macular degeneration” OR “AMD” OR “geographic atrophy”) AND “complement inhibitor”. All original papers written in English, for which full text could be obtained, were included in the initial screening. A manual search was conducted in the references of the relevant reviews with the aim of identifying further relevant studies. The systematic review was not previously registered.

### 2.1. Study Selection

The inclusion criteria were as follows: (1) clinical trials including human subjects with geographic atrophy (GA) due to AMD treated with a complement inhibitor agent; (2) a clear definition of the AMD type and area of the GA at baseline; (3) a comprehensive description of study protocol and outcomes; (4) comparison with a placebo group; and (5) a minimum follow-up period of 52 weeks. According to the PRISMA guidelines, the PICOS (participants, intervention, comparability, outcomes, study design) strategy was used to select the studies included in the final analysis:P—patients aged 50 years or over with GA due to dry AMD.I—a complement inhibitor agent was administered (any type, dose, and method of administration).C—comparison with GA and dry AMD in the placebo group.O—differences in rate of change between study groups and sham groups in terms of structural and/or functional parameters in a follow-up period of at least 52 weeks.S—prospective, interventional, and controlled trials were included.

Experimental and animal studies, meeting abstracts, case reports, and letters to the editor were excluded from the present review. Studies including patients with neovascular AMD or previously treated with anti-VEGF were not included.

### 2.2. Data Collection and Quality Appraisal of the Studies Included in the Review

A PRISMA flowchart was employed for the screening and selection of the studies included in the review. The screening was performed by two independent senior ophthalmologists with experience in AMD evaluation. Any divergence was solved through consensus. The Modified Newcastle–Ottawa Quality Assessment Scale (NOS) was employed to assess the quality of the studies and the risk of bias [24]. Only studies with a score of ≥7 were included in the qualitative analysis (Table 1).

Outcome measures evaluated included change in best-corrected visual acuity (BCVA), low luminance deficit (LLD), progression in the untransformed and square-root-transformed geographic atrophy (GA) area size, and changes in the surrounding retina. The primary structural outcome was a change in the area of geographic atrophy, measured by fundus autofluorescence (FAF) and confocal scanning laser ophthalmoscopy (CSLO) or optical coherence tomography (SD-OCT). Studies providing additional information regarding the changes in photoreceptors and retinal pigment epithelium layers in the areas surrounding geographic atrophy were included in the qualitative analysis. However, the comparability of the results was limited for these studies due to the different techniques of image acquisition and segmentation, and this could represent a possible source of bias. Additionally, while the case definition and the size of the geographic area at the inclusion were the same in all reviewed studies, the baseline evaluation did not include in all cases information regarding the presence of the specific risk factors and the nonexudative neovascular membranes in the study eye.

The safety of the complement inhibitor therapy was assessed in terms of serious systemic and ocular adverse effects (SAE), persistent increased IOP ≥ 30 mmHg, development of macular choroidal neovascularization (CNV) or exudative AMD, development of endophthalmitis, and ocular inflammation.

#### Data Analysis

Data analysis was performed using MedCalc Software© (version 23.0.2, 2024 MedCalc Software Ltd., Ostend, Belgium). When multiple studies published results referring to the same enrolled groups of patients, only the study with the most complete data was included in the quantitative analysis; this was intended to avoid bias related to duplicates. A comparison of means was performed using a t-test. A meta-analysis for continuous measures was performed using Hedges g statistic (MedCalc Software©) for the standardized mean difference (SMD) and a confidence interval (CI) for the relevant outcomes. The interpretation of the SMD was performed according to Cohen’s rule of thumb, with a value of 0.2 indicating a small effect, a value of 0.5 indicating a medium effect, and a value of 0.8 or more indicating a larger effect. The heterogenicity of the studies was analyzed using Cochran’s Q test and the I^2^ test for inconsistency. If *p* < 0.1 and I^2^ > 50%, the heterogeneity was considered high, and a random-effects model was used to report data for the meta-analysis, with a more conservative approach and a wider CI for estimated effects.

## 3. Results

The review was carried out on papers published in the last decade, from 2014 to 2024. After duplicate removal and the application of the inclusion and exclusion criteria and reviewing the full text, 18 studies published between 2014 and 2024 were included in the qualitative analysis (Figure 1).

The complement inhibitors analyzed for potential therapeutic effects were as follows: C5 was studied in [25,26,40,41,42]; complement factor D was studied in [27,28,39]; C3 was the focus of [29,30,31,32,33,34,35,36,37,38,42]. The pharmacologic agent was administered through the intravenous route in 2 (11%) studies [25,26], and it was administered through intravitreal injections in the other 16 (94.5%).

All studies were prospective, randomized control trials, with a follow-up time varying from 52 to 104 weeks, with a mean of 68.2 ± 20.4 weeks. The same primary endpoint was considered by most studies in testing the efficacy of complement inhibitors for GA in AMD (10, 55.5%): a 15% decrease in GA area growth at the end of the follow-up period between treated arms and sham [25,27,28,29,30,32,36,38,40,41]. GA area was evaluated using FAF in nine studies (50%); it was evaluated by combining fundus autofluorescence (FAF) with other objective structural evaluation methods, such as confocal scanning laser ophthalmoscopy (CSLO) or spectral domain optical coherence tomography (SD-OCT), in four studies (22.2%). The functional evaluation of the effects of complement inhibitor administration was evaluated in 9 [25,27,28,29,30,32,36,38,40,41] out of 18 studies (50%). This analysis consisted of normal-light best-corrected visual acuity (NL-BCVA) (9/9) and low-luminance BCVA (LL-BCVA) while extracting the low-luminance deficit (LLD) in seven out of nine studies. The relevant data abstracted from these studies are presented in Table 2.

The COMPLETE study [25,26] found that intravenous eculizumab proved to be associated with no functional or structural improvement in patients with GA and AMD, compared to the natural history of the disease [25,26]; in comparison, the intravitreal route of different complement inhibitor agents was associated with various degrees of improvement in anatomical outcomes compared to the natural course of the disease. Moreover, given the imbalance between the COMPLETE study and the other studies in terms of baseline characteristics (BCVA and lesion size), which largely explain the absence of a therapeutic effect of eculizumab (Table 3), we decided to exclude it from further analysis.

### 3.1. Demographic and Inclusion Criteria of the Patients in the Quantitative Analysis

The total number of the patients included in the studies included in the quantitative analysis [27,28,29,38,40,41] was 4160, out of which 2684 (64.5%) were treated with different complement inhibitor agents. The mean age varied little among the study groups from 77 [41] to 80.9 years [29], with a pooled mean value of 78.1 ± 7.8 years for the treated arms and 77.8 ± 7.9 years in the placebo group (*p* = 0.67). Most of the patients were females (2644, 63.5%), varying from 43.9% [27] to 72.6% [32] among the study subgroups. When reported, the majority of the patients were white (3836, 92.2%; Table 4).

The general inclusion criteria were comparable among the analyzed papers. In all cases, patients were aged over 50 years of age, diagnosed with AMD and GA (based on fundus autofluorescence images (FAF)), with a total GA area at baseline varying between 2.5 and 17.5 mm^2^ (1- and 7-disc areas) or multifocal GA lesions, with at least one focal lesion ≥ 1.25 mm^2^ (0.5-disc area).

However, there were differences in the patient selection procedures that could possibly explain the differences in the outcomes. There were no specific criteria regarding the excentricity of GA to the fovea, except for GATHER 1 and GATHER 2 [40,41], in which the GA lesion had to be the non-center point, usually within 1.5 mm of the foveal center.

BCVA was not an inclusion criterion in the MAHALO study [27]. In the CROMA and SPECTRI studies, for the studied eyes with BCVA letter scores of 79 or more (Snellen equivalent, 20/25 or better), at least 1 lesion was required within 250 μm of the foveal center. In the FILLY [29], DERBY, and OAKS studies [38], the only requirement was the presence of a BCVA of 24 letters or more as assessed using the Early-Treatment Diabetic Retinopathy Study Charts. For GATHER 1 and 2, a BCVA between 20/25 and 20/320 in the study eye was an eligibility criterion.

CNV in the fellow eye was an exclusion criterion in three (16.6%) of the studies [28,32,41], while this was allowed in other studies. The incidence of neovascular AMD at baseline was reported only by Liao et al. [29], varying from 35.4% to 41.9 among the study subgroups.

The baseline mean NL-BCVA was comparable between the treated and sham groups (combined mean 63.7 ± 13.4 letters vs. 64.2 ± 13.7 letters, *p* = 0.24), ranging between 80 and 25 ETDRS letters in all studies, meaning a Snellen equivalent to 20/25–20/320 (see Tabel 4).

### 3.2. Functional Outcomes of the Complement Inhibitor Therapy for GA in AMD

NL-BCVA varied widely for both the treated groups (0.7; −8.8 letters) and the sham groups (+2.9; −9.3 letters). However, the pooled mean changes at the end of the follow-up period were similar in the treated groups vs. the sham groups for both NL-BCVA scores (−5.5 letters vs. −5.4 letters, *p* = 0.45).

The meta-analysis for the continuous measure was performed using MedCalc Software© (2024 MedCalc Software Ltd., Ostend, Belgium). The results showed significant heterogeneity in the changes in NL-BCVA among the studies (Q test, *p* < 0.0001; I^2^ for inconsistency, 99.7%).

The random-effect model was chosen for interpretation due to the high heterogeneity among studies, which showed no significant change between the treated groups and the placebo groups (standardized mean difference: −0.213; CI: −2.29 to 1.86; *p* = 0.84). The risk of bias was not assessed due to the limited reliability of Egger’s and Begg’s tests if fewer than 10 studies are included in a meta-analysis [43,44] (Table 5; Figure 2).

The high heterogeneity among studies may be explained by multiple reasons: the large window for BCVA used as an inclusion criterion, the different follow-up periods among studies, the potentially different effects of therapeutic agents on BCVA, the location of the initial GA lesions, and the directions of growth. There is little correlation between VA and GA due to AMD, with most patients preserving good vision if the fovea is spared, regardless of the size of the geographic atrophy. For this reason, BCVA was not considered a primary outcome in the reviewed studies.

A comprehensive functional evaluation—which could be more appropriate for patient follow-up for GA in AMD—was performed by Heier et al. [38] in the OAKS and DERBY trials. The evaluation included the change in the monocular speed read using the MNREAD/Radner chart, the mean functional reading independence index score, and the changes in the mean threshold sensitivity using microperimetry. None of the studies could find a significant difference between the treated and sham arms in any functional tests.

### 3.3. Structural Outcomes of the Complement Inhibitor Therapy for GA in AMD

As the primary outcome, all studies included in the present meta-analysis focused on structural changes and investigated the comparative increase in the GA area between the treated arms and the shams. A minimum of 15% decrease in growth was set as an endpoint for complement inhibitor efficacy. Applying the square root transformation to GA lesion area change was preferred by some authors in reporting outcomes [25,28,29,32,40,41]; this is because it has been shown to eliminate the dependence of the GA lesion growth rate on the baseline area [29] as a possible confounder.

The intravitreal administration of complement inhibitors was associated with various degrees of decrease in the growth of the GA area compared to the placebo (Figure 3).

The primary endpoint—a 15% decrease in GA growth—was reached in four out of six studies [27,29,40,41]; it was partially reached in the research published by Heier et al. [38] (for the OAKS study). Despite the encouraging results presented by the MAHALO study on lampalizumab, no significant therapeutic effects were observed in either study arm in the phase 3 SPECTRI and CROMA studies.

Studies involving both pegcetacoplan and avacincaptad pegol [29,38,40,41] found that the therapeutic effect became evident after 6 months in a dose-dependent manner.

For pegcetacoplan, stronger effects in decreased GA area growth were observed in the monthly-treated patients compared to those who underwent intravitreal injections every other month [29,38]. However, evaluation after 12 months from the initiation of therapy showed differences in effects among studies, from 29% vs. 20% in the FILLY study [29] to 21% vs. 16% in the OAKS study to 12 vs. 11% in the DERBY study [38]. Interestingly, the differences were less important at the 24-month follow-up (22% vs. 18% decrease in GA growth in OAKS and 19% vs. 16% decrease in DERBY study). A total of 729 patients in the DERBY and OAKS studies continued the therapy; patients were evaluated at 30 months in the GALE study [44]. The investigators found a 24% decreased growth in monthly arms and a 21% decrease in the PEOM group. Compared to the linear progression that was observed in the sham group, the slopes of the progression continued to decrease over time for those who were treated. In addition, there was a divergent effect—at every 6-month interval, the medication had a greater effect [45].

Studies involving avacincaptad pegol found comparative effects for 2 mg vs. 4 mg per dose for monthly treatments [32,40,41]. In the second year of the GATHER 2 study, the treated patients were re-randomized in a 1:1 manner in the monthly-treated arm and the every-other-month-treated arm. Recent data from the 24-month follow-up showed similar effects in the two study arms compared to the sham patients (14% and 19% reduction, respectively), with a more than doubled effect in 2 years of treatment compared to 1 year of treatment [46].

Furthermore, we conducted a meta-analysis for continuous measures to investigate the effect of complement inhibitor agents in the reviewed studies on the square root transformation to the GA lesion area. If there were multiple treated arms in the study, then the one with the best outcome was selected when conducting the comparison with the placebo. The random-effects model showed a significant effect of the complement inhibitors associated with decreased growth of GA (*p* < 0.001). When the pooled results were analyzed, pegcetacoplan and avacincaptad pegol were shown to have stronger effects than lampalizumab (Table 6; Figure 4).

However, the results might be affected by the potential risk of bias. We observed a very high heterogeneity (Q: 951.08; DF: 6; *p* < 0.0001; I^2^: 99.37%), which could be due to several factors, such as differences in inclusion criteria, follow-up period, sample size, or random effects. A higher effect was observed in phase 2 studies compared with phase 3 investigations on the same therapeutic agents; this is probably due to the small-study effect [44]. The rate of growth varied significantly among studies in both the treated and the sham groups (Table 6). GA in AMD is a heterogenous condition, with the rate and direction of progression being variable among subjects. Studies involving pegcetacoplan included both non-subfoveolar GA lesions and subfoveolar lesions, while those using avacincaptad pegol included only non-subfoveolar lesions.

Moreover, genetic variance among individuals in the treated arms could play a significant role. In the MAHALO study [27], the efficacy of lampalizumab significantly increased in the CFI risk allele carrier subgroup (44% vs. 20% reported in the entire group). However, genetic analysis was not performed by other studies.

### 3.4. Other Imagistic Biomarkers

Furthermore, with the wide availability of SD-OCT in ophthalmological practice, other OCT-based biomarkers were investigated to document the effects on the photoreceptors layer and the RPE in the areas surrounding geographic atrophy. Several post hoc studies on pegcetacoplan found a delay in PR and RPE degeneration situated in the junctional zone, an area of a 500 µm band surrounding the GA in the treated arms versus the sham [33,34,36,42]. The OCT-based evaluation found significantly less thinning in all three laminae corresponding to the photoreceptor: the outer nuclear layer (ONL) and the photoreceptor inner and outer segments (IS and OS) [34]. Riedl et al. found that pegcetacoplan seems to have a potentially higher effect in preserving PR than RPE [37]. In the study of Nytalla et al. [33], the progression from incomplete to complete retinal pigment epithelium and outer retina atrophy (RORA) outside 500 µm surrounding GA was decreased by 48% in the monthly-treated group and by 24% in the EOM group compared to controls.

### 3.5. Factors Correlated with GA Progression and Response to Therapy

A nonuniform pattern of GA progression was described in the reviewed studies. Foveal involvement was a key predictor of geographic atrophy lesion growth [38]. The progression of GA was correlated with higher LLD at baseline [30], extrafoveal lesions [30,32,35], progression direction, photoreceptor (PR) integrity, and hyperreflective foci status (HRF) [35]. On the other hand, in a large cohort study, Heier et al. found that the complement inhibitor effect was higher in non-subfoveal GA vs. subfoveal GA [38].

### 3.6. Genetic Analysis

Extensive genetic analysis for the at-risk allele involving multiple genetic variants, including CFH, CFI, C2/CFB, and C3 was tested in four studies [25,27,28,29]. Yaspan et al. [27] found that carriers of the CFI risk allele were associated with GA progression and response to intravitreal Lampalizumab, while rs2230199 in C3 and rs3750846 in ARMS2 were correlated with GA growth, independently of the treatment arm, in the study conducted by Liao et al. [29]. However, no specific correlations were found with the response to complement inhibitor therapy.

### 3.7. Safety of Intravitreal Complement Inhibitors

Complement inhibitors proved to have a good safety profile for both intravenous and intravitreal administration. In the reviewed studies, the systemic adverse events were similar in both the treated and the sham groups and were more correlated with advanced age than with the complement inhibition effect.

The ocular adverse effects were comparable to those of anti-VEGF injections. The most-reported adverse effects were conjunctival hemorrhage, conjunctival hyperemia, dry eye, eye pain, punctate keratitis, increased intraocular pressure, macular neovascularization, vitreous detachment, and cataracts [41]. Increased intraocular pressure (IOP) over 30 mmHg was observed in 1.3–8.3% of cases, but it was not persistent and could be managed with topical and/or general medication or anterior chamber paracentesis. Infectious and sterile endophthalmitis was reported for intravitreal lampalizumab and pegcetacoplan, but the incidence was low, varying between 0.04% [28] and 1–2.3% [29,38], respectively. See in Table 7.

One specific concern was raised by the development of exudative AMD and CNV in the eyes of the studied participants. A total of 138 (5%) cases were reported in the treated arms, and 30 (1.9%) cases were reported in the sham group (*p* < 0.001). The effect was reported in intravitreal but not intravenous administration, with an incidence varying from 1.1% [28] to 20.9% [29]. These differences might be explained in part by the differences in the criteria and baseline examination of the patients. For some studies, the presence of exudative AMD in either eye was an exclusion criterion [28,32,41], while in others, exudative AMD in the fellow eye was admitted [29,38]. Moreover, OCT examination at baseline was not available in all centers, so nonexudative macular neovascular membranes could be missed at initial screening.

A post hoc analysis of patients included in the FILLY trial [31] found that the risk factors for developing exudative AMD during the follow-up period were reported to be the presence of CNV at inclusion in the fellow eye and the presence of double sign layer (DLS), which is suggestive of nonexudative MNV at the baseline in the treated eye. The effect appeared between 31 and 255 days of treatment in the study group, with no specific cluster in time, but the effect occurred in a dose-dependent manner [31]. In all cases, the patients responded well to anti-VEGF therapy, and visual acuity could be preserved. Jaffe et al. [32] suggested that the conversion of nonexudative MNV to exudative MNV in these eyes may have an immunologic basis; however, the exact pathological mechanism could not be elucidated.

## 4. Discussion

### 4.1. Complement System in AMD Pathology and Potential Limitations of Complement Inhibitor Therapy

One of the main limitations of our meta-analysis was the significant heterogeneity encountered in the reviewed studies in terms of preventing disease progression. Although the inclusion criteria were similar—and in all studies except the COMPLETE study [25], the agent was administered intravitreally—taking a comparative approach, the main cause may be related to the specific inhibitory effects of the treatments.

Complement activation may be achieved by either classical and lectin involving C4, or by an alternate pathway through C3 thick-over and spontaneous hydrolysis (Figure 5).

The complement inhibitors in the reviewed studies act at different levels of the complement activation cascade, each with its own advantages and limitations. Only one study [42] involving lampalizumab investigated the changes in complement fractions in aqueous humor after administrating the complement inhibitor. Lampalizumab is a specific inhibitor of CFD, which is involved in the alternate pathway through cleaving factor B. Edmons et al. found that intravitreal administration of lampalizumab increased CFD levels significantly by 92–124%, through efficiently binding with the therapeutic agent; it also decreased the Bb–CFB ratio by 41–43%, proving its efficacy in preventing the cleavage of CFB to Bb [42]. However, lampalizumab had no effect on C3, C3 processing, C4, or C4 processing. Inhibiting only the alternative pathways via factor D had a limited effect in preventing the progression of GA.

Pegcetacoplan binds C3 and prevents its cleavage, suppressing the downstream effects of excessive complement activation at the level of C3 and simultaneously inhibiting the classical, lectin, and alternative pathways [29,30,34]. While no particular genetic variants associated with AMD were related to C5, it was considered a prime target in blocking the formation of MAC complexes [14,47]. A further downstream blockage at the C5 level by avacincaptad was associated with earlier clinical results by some authors. However, the limitations were considered to be related to uncontrolled chronic depositions of C3 fragments at the level of targeted cells, exhibiting proinflammatory effects [48,49].

### 4.2. Current Challenges in Complement Inhibitor Therapy for GA Due to AMD

Age-related factors were reported to be one of the categories affected by delays and disruptions in diagnosis and therapy during the COVID-19 pandemic, with detrimental effects on visual outcomes in AMD patients [50,51,52,53]. There is still a high unmet need for novel therapeutic strategies in the management of AMD. While multiple factors were proved to be involved, there is solid evidence that the hyperactivation of the complement system is associated with photoreceptor damage and atrophy of RPE. The reviewed studies added solid evidence that the complement system is involved in AMD pathology and progression and may represent a therapeutic target in geographic atrophy.

However, functional parameters had limited efficacy in monitoring the results; in other words, patients should not expect a significant visual improvement [54,55]. The main explanation is that there is little correlation between the growth of GA and visual function. The location of GA, being closer to the foveal region rather than the size, is moreover correlated with a significant decrease in BCVA. Moreover, a study using adaptative optics [56] found that VA is a low-sensitivity tool in measuring foveal cone damage until the cone spacing reaches up to 40% of normal. Due to its most common site of onset being outside the fovea, GA may be associated with long-term good vision. Testing of BCVA may inadequately assess functional impairment in individuals with preserved foveal function despite loss of pericentral macula. On the other hand, geographic atrophy is the irreversible end-stage for the involved area, and once vision is impaired by foveal atrophic changes, it cannot be restored, even if the therapy is efficient in preventing progression [37].

Other measures, including fundus-controlled microperimetry, reading speed, and patient-reported outcomes related to quality of life, might be worthwhile investigating in efforts to improve impairments of visual function that occur for patients with GA [28,29,54,55]. However, these are time-consuming approaches and are not currently used in clinical practice. Among the reviewed studies, only Holz et al. performed a comprehensive functional evaluation using these measures, but the study found no significant changes in either structural or functional tests [28].

In terms of the anatomical results, the patients should expect a decreased rate of progression over time. The therapeutic effect might be seen after at least 6 months of treatment and may be influenced by multiple individual factors. Monthly dosing may lead to optimal outcomes, especially in the first year of treatment, but will not be realistic in terms of patient adherence. Moreover, a slightly higher efficacy with monthly treatment may be associated with higher risks of CNV, ischemic optic neuropathy, ocular infections, and an increased treatment burden for patients and their families.

The safety of intravitreal complement inhibitor therapy is a crucial aspect, given the need for long-term treatment with frequent injections. Intravitreal injections are procedures that have possible associations with endophthalmitis, regardless of the therapeutic agent used. Complement inhibitors may favor this complication due to the suppression effect in immune defense mechanisms. However, endophthalmitis incidence was low in reported studies, varying from 0.4% for intravitreal lampalizumab and 1.7% for pegcetacoplan [28,38,57]; it did not occur with avacincaptad pegol [40,41]. This fact may be related to the lower number of subjects included for the latter agent, but it may also be a result of the mechanism of the agent and the level of complement inhibition exerted. C5 inhibition might have a potentially lower risk of endophthalmitis, preserving C3-associated effector functions, such as C3b-mediated opsonization and immune cell activation, compared to an upstream C3 blockade, which also abrogates C3b effects [14,38,58,59].

Another subject of concern is the higher risk of nAMD in patients treated with complement inhibitors compared to sham. The previously published studies found a 3–9-fold increase in incidence for pegcetacoplan, and a three- to fourfold increase for avacincaptad pegol over 2 years of follow-up [29,38]. While these findings may be partially explained by including patients with nAMD in the fellow eye in some studies or undiagnosed fine nonexudative CNV in subgroups where OCT examination was not available [31], further research is needed to understand these findings. Further development of new therapeutic agents, combining anti-VEGF and complement inhibition, could help in managing this complication.

### 4.3. Future Directions of Research

#### 4.3.1. Complement Inhibitors in Neovascular AMD

Undergoing studies are investigating complement inhibitor therapy in association with anti-VEGF in neovascular AMD, and they are finding promising results [14,47]. Efdamrofusp alfa is a novel bispecific molecule that acts both as an anti-VEGF agent and a C3/C4 inhibitor; it is currently being used in an ongoing phase 3 study [47]. It has high binding affinities for VEGF165, VEGF121, and placental growth factor (PIGF), similar to that of aflibercept and higher than that of bevacizumab [14]. Previous experimental studies have shown a high affinity for binding to C3b and C4b, leading to the simultaneous inhibition of the classical and alternative pathways, possibly surpassing the limitations of the previously studied complement inhibitors.

#### 4.3.2. Novel Biomarkers and a Personalized Therapeutic Approach

Implementing complement inhibitor therapy in clinical practice may be challenging, considering the significant burden of frequent intravitreal injections and the less-understood effect of possible choroidal neovascularization [60,61]. Adherence to therapy and increased costs, which may be close to USD 2000/injection [62], are important to consider. The therapy is well tolerated and may be safely performed on a daily basis [32,38,41,63]. However, informed consent should be obtained after a reasonable presentation of associated risks and benefits based on a personalized approach [64,65].

Patients with AMD present a heterogeneous rate of progression varying from 0.4 to 53% [66]. The decision of whether to treat or not, and how aggressive the management plan should be, should be based on a personalized approach [67,68]. Prediction of the rate and the pattern of growth in an individual is extremely important in estimating the imminent threat to the fovea and, with it, finely detailed central vision [63,64,68]. As evidenced by the variability of the changes in lesion size and the absence of functional advantages, other effect markers—such as PR and RPE loss, EZ impairment, and progression from iRORA to cRORA—may be more important in individualizing treatment and predicting long-term functional outcomes. These new biomarkers might be more useful for defining and controlling the success of treatment. The degree of photoreceptor degeneration, assessed by specific laminae thinning, may offer earlier endpoints in patient selection and monitoring [34,35]. Understanding the progression of lesions in atrophic AMD requires the development of new biomarkers based on SD-OCT analysis and AI monitoring models.

## 5. Conclusions

Complement inhibitors are promising therapeutic agents for GA in AMD. Intravitreal administration decreased the rate of GA growth and exhibited a protective effect on photoreceptors and RPE in the adjacent area. However, the burden of injections and the risk of nAMD must be weighed in relation to the risk–benefit balance. A better selection of cases relying on personalized management could improve outcomes and increase the chances of this therapy being implemented in ophthalmology practice.

## Figures and Tables

**Figure 1 jpm-14-00990-f001:**
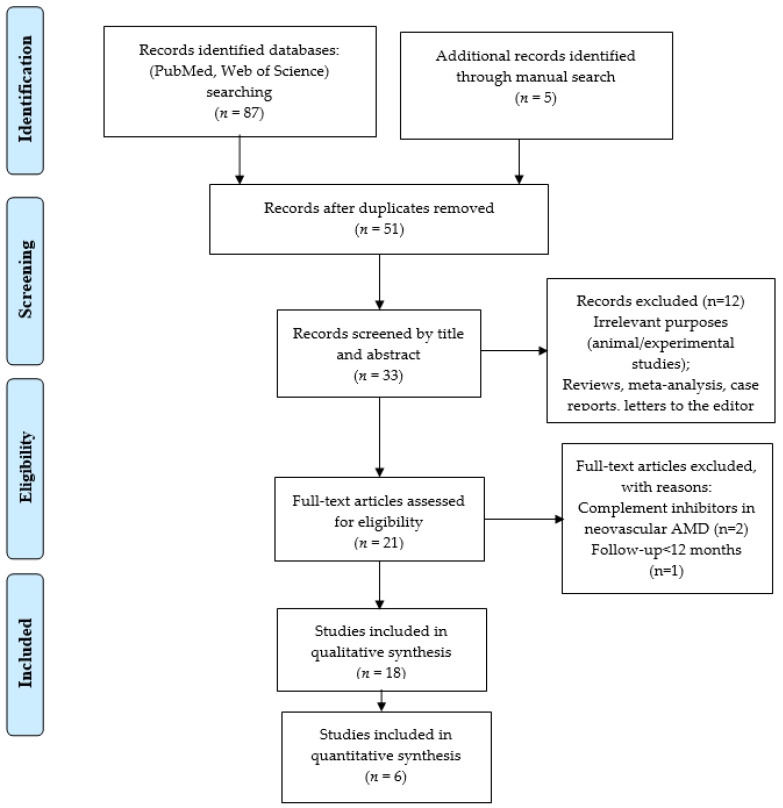
PRISMA flowchart for the studies included in the review.

**Figure 2 jpm-14-00990-f002:**
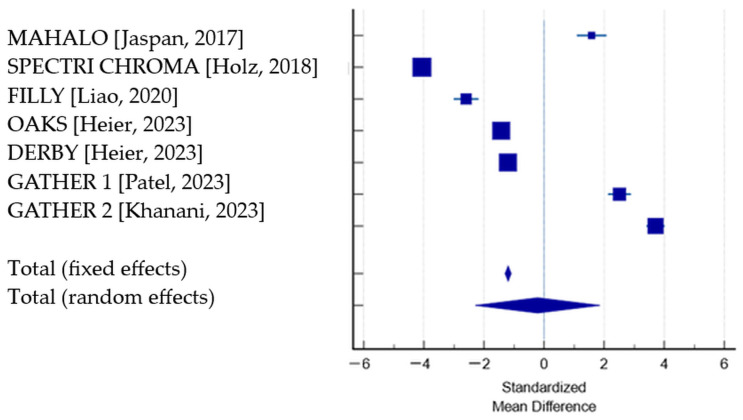
Effects of complement inhibitors vs. sham on BCVA change: Forrest plot—pooled effects–random effects model (MedCalc Software©). References [27,28,29,38,40,41].

**Figure 3 jpm-14-00990-f003:**
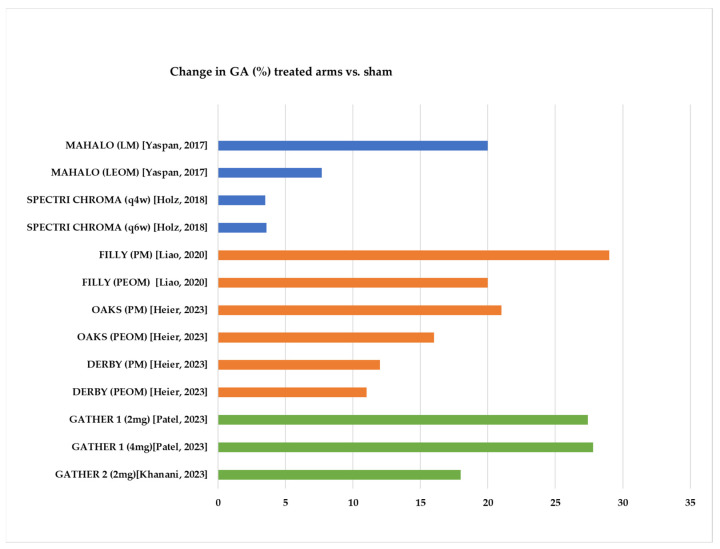
Differences in the growth of the GA area during the follow-up period between the treated arms and the controls [26,28,29,38,40,41]. Color code: blue—lampalizumab; orange—pegcetacoplan; green—avacincaptad pegol; LM—lampalizumab monthly; LEOM—lampalizumab every other month; q4w—every 4 weeks; q6w—every 6 weeks; PM—pegcetacoplan monthly; PEOM—pegcetacoplan every other month.

**Figure 4 jpm-14-00990-f004:**
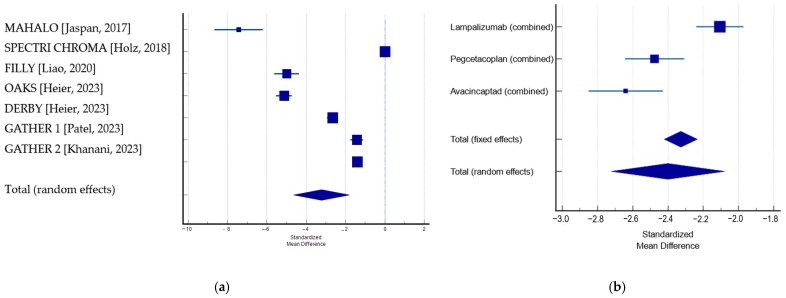
Effects of intravitreal complement inhibitors vs. sham on GA area change. Forrest plots—pooled effects–random effects models: (**a**) for studies included in the meta-analysis [27,28,29,38,40,41]; (**b**) for pool results of the studies involving lampalizumab, pegcetacoplan, and avacincaptad pegol.

**Figure 5 jpm-14-00990-f005:**
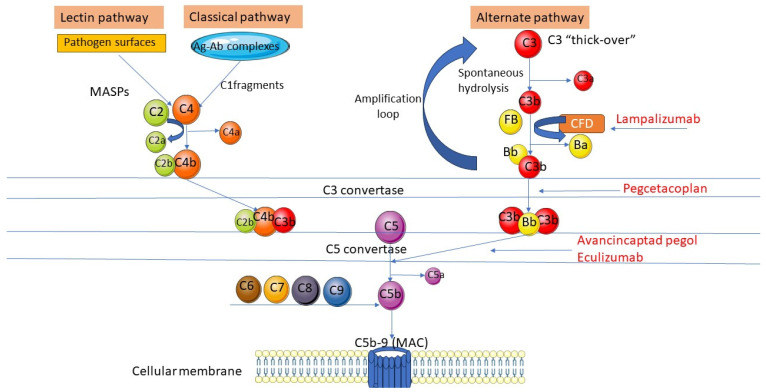
Targeting complement cascade for GA in AMD.

**Table 1 jpm-14-00990-t001:** Analysis of risk of bias using the Modified Newcastle–Ottawa Scale (NOS) for the studies included in the review.

	Yehoshua et al., 2014 [25]	Garcia Filho et al., 2014 [26]	Yaspan et al., 2017 [27]	Holz et al., 2018 [28]	Liao et al., 2020 [29]	Steinle et al., 2021 [30]	Wykoff et al., 2021 [31]	Jaffe et al., 2021 [32]	Nittalla et al., 2022 [33]	Pfau et al., 2022 [34]	Vogl et al., 2023 [35]	Mai et al., 2022 [36]	Riedl et al., 2022 [37]	Heier et al., 2023 [38]	Edmonds et al., 2023 [39]	Patel et al., 2023 [40]	Khanani et al., 2023 [41]	Fu et al., 2024 [42]
Selection (≤4)	3	3	4	4	4	4	4	4	4	4	4	4	4	4	3	4	4	4
Adequate case definition	★	★	★	★	★	★	★	★	★	★	★	★	★	★	★	★	★	★
Representativeness of cases	?	?	★	★	★	★	★	★	★	★	★	★	★	★	★	★	★	★
Selection of controls	★	★	★	★	★	★	★	★	★	★	★	★	★	★	?	★	★	★
Control definition	★	★	★	★	★	★	★	★	★	★	★	★	★	★	★	★	★	★
Comparability (≤2)	2	2	2	2	2	2	2	2	2	2	2	2	2	2	2	2	2	2
Considers the effect of age	★	★	★	★	★	★	★	★	★	★	★	★	★	★	★	★	★	★
Considers the effect of any additional factor	★	★	★	★	★	★	★	★	★	★	★	★	★	★	★	★	★	★
Outcome (≤3)	3	3	2	2	3	2	2	3	2	2	2	2	2	3	1	3	3	2
Objective, validated laboratory method	★	★	★	★	★	?	?	★	?	?	?	?	?	★	?	★	★	★
Blinded assessment	★	★	★	★	★	★	★	★	★	★	★	★	★	★	?	★	★	?
Appropriate statistical analysis	★	★	★	★	★	★	★	★	★	★	★	★	★	★	★	★	★	★
Total Score (≤9)	8	8	9	9	9	8	8	9	8	8	8	8	8	9	7	9	9	8

Footnote: ★ the question is addressed in the study; ? not identified in the study; Green shading denotes a low risk of bias as defined by an NOS score of 7 or greater.

**Table 2 jpm-14-00990-t002:** Studies included in the review using complement inhibitors for geographic atrophy (GA) in age-related macular degeneration (AMD).

Study, Year	Agent, Dose	Action	Type, Design	No. Patients	Follow-Up (w)	Primary Outcomes	Secondary Outcomes	Outcomes
Yehoshua Z [25], 2014	iv Eculizumab 600 mg/w (low dose)/900 mg/w (high dose), 4 w, then 900 mg/w (low dose)/1200 mg/w (high dose) every 2 w until w24	C5 inhibitor	COMPLETE study, phase 2, RCT	30 (low-dose group; high-dose group; placebo: 10;10;10)	52 w	Change in mean GA area (CSLO and FAF)	Change in BCVA and LLD (ETDRS)	No difference in structural and functional outcomes at w26 and w52 between treated groups and controls; initial LLD correlates with GA progression at 6 months
Garcia Filho CA [26], 2014	iv Eculizumab 600 mg/w (low dose)/900 mg/w (high dose), 4 w, then 900 mg/w (low dose)/1200 mg/w (high dose) every 2 w until w24	C5 inhibitor	COMPLETE study, phase 2, RCT	30 (low-dose group; high-dose group; placebo: 10;10;10)	52 w	Change in mean square root Drusen volume in central 3 mm and 5 mm area measured using SD-OCT	-	No differences between treated groups and placebo at 26 w and 52 w (0.02 vs. −0.05, *p* = 0.17)
Yaspan BL [27], 2017	Intravitreal lampalizumab, 10 mg	Factor D inhibitor	MAHALO phase 2 study (42 L M; 41 LEOM, 40 SM)	123	78 w	Change in mean GA area (FAF)	Change in BCVA (ETDRS)	Reduction rate of GA area change: 20% ↓ LM vs. SM; 7.7% ↓ LEOM vs. SM (no effect)
Holz FG [28], 2018	Intravitreal lampalizumab, 10 mg	Factor D inhibitor	CHROMA (906) and SPECTRI (975) phase 3, RCT (628 q4w; 627 q6w; 626 SM)	1881	96 w	Change in mean GA area (FAF)	Change in BCVA and LLD (ETDRS)	No effect in treating GA (↓ by −3.6; −3.5 vs. SM)
Liao D [29], 2020	Intravitreal pegcetacoplan (APL-2) 15 mg (0.1 mL)	C3 inhibitor	FILLY trial, phase 2, RCT	246 (84: PM; 78: PEOM; 80: SM)	78 w	Square root of the change in the GA area	Change in BCVA and LLD (ETDRS); untransformed GA area; foveal encroachment (FAF)	↓ Growth of square root GA in treated arms (29% PM vs. SM; 20% PEOM vs. SM); 30% and 20% ↓ in untransformed GA area growth at month 12 in the PM and PEOM vs. SM; no effect on changes in foveal encroachment, BCVA, LLD
Steinle N [30], 2021	Intravitreal Pegcetacoplan (APL-2) 15 mg (0.1 mL)	C3 inhibitor	post hoc, FILLY trial	192 (67 PM; 58 PEOM; 67 SM)	52 w	Risk factors for GA progression		Extrafoveal lesions and larger LLD: potential risk factors for GA progression; ↓ GA progression in treated groups after adjusting these risk factors
Wykoff CC [31], 2021	Intravitreal Pegcetacoplan (APL-2) 15 mg (0.1 mL)	C3 inhibitor	post hoc, FILLY trial	246 (26 with new eAMD)	78 w	Risk factors for new-onset exudative AMD (eAMD) in the study eye		Baseline eAMD (18; 69%) in the fellow eye and DLS in the study eye (19; 73.1%) are associated with new eAMD, in a dose-dependent manner
Jaffe GJ [32], 2021	Intravitreal avacincaptad pegol 2 mg (0.1 mL)/4 mg (2 injections of 0.1 mL)	C5 inhibitor	GATHER 1 phase 2/3	286, two phases; 67 (2 mg); 110 (sham); phase 2: 85 (4 mg); 86 (sham)	52 w	Mean rate of change in GA (FAF and SD-OCT)	Change in NL and LL-BCVA (ETDRS)	↓ Growth in GA area by 27.4% (2 mg group vs. sham) and 27.8% (% (4 mg group vs. sham); no difference for NL BCVA and LLD
Nittala MG [33], 2022	Intravitreal pegcetacoplan (APL-2) 15 mg (0.1 mL)	C3 inhibitor	post hoc, FILLY trial	167 (41 PM; 56 PEOM; 70 SM)	52 w	Progression from iRORA to cRORA outside 500 µm surrounding GA, using the Zeiss Cirrus/Heidelberg Spectralis OCT	-	Risk of progression to cRORA in treated arms vs. sham: 48% ↓ (PM) and 24% ↓ (PEOM)
Pfau M [34], 2022	Intravitreal Pegcetacoplan (APL-2) 15 mg (0.1 mL)	C3 inhibitor	post hoc, FILLY trial	192 (67 PM; 61 PEOM; 64 SM)	78 w	PRD along the 5.16° and 2.58° contour-line of GA by Heidelberg Spectralis OCT	-	ONL, IS, and OS thickness ↑ in treated groups vs. control (*p* < 0.001); the effect is ↑ in PM vs. PEOM
Vogl WD [35], 2022	Intravitreal pegcetacoplan (APL-2) 15 mg (0.1 mL)	C3 inhibitor	post hoc, FILLY trial	146 (57 PM; 46 PEOM; 53 SM)	52 w	AI-based GAMM fit for LPR	-	Highly nonuniform LPR related to eccentricity to the fovea, progression direction, PR integrity, and HRF; when controlling co-factors, LPR ↓ by −28.0% (monthly) and −23.9% (EOM) vs. sham
Mai J [36], 2022	Intravitreal pegcetacoplan (APL-2) 15 mg (0.1 mL)	C3 inhibitor	post hoc, FILLY trial	113 patients (38 PM, 36 PEOM, and 39 SM)	52 w	Correlation of GA (FAF vs. SD-OCT); characterization of other OCT biomarkers	-	Excellent agreement in GA area evaluation; ↓ RPE atrophy progression and ↓ in EZ impairment (SD-OCT) in treated groups, potentially more sensitive in monitoring GA therapy
Riedl S [37], 2022	Intravitreal pegcetacoplan (APL-2) 15 mg (0.1 mL)	C3 inhibitor	post hoc, FILLY trial	162 (52: PM; 54: PEOM; 56: SM)	52 w	Inhibition of PR loss and thinning in GA on SD-OCT by deep-learning-based automated PR quantification	-	PR loss and thinning ↓ in treated arms; higher ↓ in the PR loss/RPE loss ratio in PM vs. SM (C3 inhibition prevents more PR loss than RPE loss)
Heier JS [38], 2023	Intravitreal pegcetacoplan (APL-2) 15 mg per 0.1 mL	C3 inhibitor	OAKS/DERBY studies, phase 3, RCT	1258 OAKS 614 (202 PM; 205 PEOM; 207 SM); DERBY 597 (201 PM; 201 PEOM; 195 SM)	104 w	Change in mean GA area (FAF)	Change in monocular maximum speed reading (MNREAD/Radner chart), mean functional reading independence index score, BVCA, LLD (ETDRS); change from baseline in the mean threshold sensitivity of all points in the study eye by mesopic microperimetry (OAKS only)	↓ in GA area change in treated vs. SM: At 12 m: –21% (PM); –16% (PEOM) for OAKS; –12% (PM); –11% (PEOM) for DERBY (endpoint not met) At 24 m: −22% (PM); −18% (PEOM) for OAKS; ↓by 19% (PM); 16% (PEOM) in DERBY The effect was higher in non-subfoveal GA vs. subfoveal GA; no difference in functional endpoints
Edmonds R [39], 2023	Intravitreal lampalizumab, 10 mg	Factor D inhibitor	CHROMA (906) and SPECTRI (975) Phase 3, RCT	97(35-q4w; 32-q6w; 30-SM)	96 w	Changes in Complement fractions in aqueous humor: CFD, full-length CFB; the Bb fragment; full-length C3; the C3c, iC3b, C3b; full-length C4; the C4c, C4b		a 92% to 124% median ↑ in CFD levels; 41% to 43% median ↓ in the Bb: CFB ratio no effect on C3, C3 processing; C4, C4 processing;
Pattel SS [40], 2023	Intravitreal avacincaptad pegol 2 mg (0.1 mL)/4 mg (2 injections of 0.1 mL)	C5 inhibitor	GATHER 1 phase 2/3	201, two phases: 67 (2 mg); 110 (SM); phase 2: 83 (4 mg); 84 (SM)	78 w	Mean rate of change in GA (FAF and SD-OCT)	Change in NL and LL-BCVA (ETDRS)	↓ growth in GA area by 28.1% (2 mg group vs. sham) and 30.0% (% (4 mg group vs. sham); no difference in NL BCVA and LLD
Khanani MA [41], 2023	Intravitreal avacincaptad pegol 2 mg (0.1 mL)	C5 inhibitor	GATHER 2 phase 3	448; (225 treated; 223 SM)	104 w	Mean rate of change in GA (FAF) at 6, 12, 24 m	Change in BCVA and LL-BCVA at 12 m (ETDRS letters)	↓ growth in GA area by 18% (2 mg group vs. sham) at 12 m; no difference in change in BCVA and LL-BCVA
Fu DJ [42], 2024	Intravitreal pegcetacoplan (APL-2) 15 mg per 0·1 mL	C3 inhibitor	post hoc for OAKS/DERBY studies	936	104 w	Change in GA area, PRD, RPE loss, RORA (SD-OCT)	Change in NL and LL-BCVA (ETDRS)	Delay in atrophy of both the RPE and PR in treated arms vs. sham, starting from month 2; no correlation with BCVA

Note: iv—intravenous; w—week; RCT—randomized control trial; FAF—fundus autofluorescent; CSLO—confocal scanning laser ophthalmoscopy; ETDRS—scale for visual acuity (measured in letters) used initially in Early Treatment of Diabetic Retinopathy Study; LM—lampalizumab monthly; LEOM—lampalizumab every other month; SM—sham; q4w—every 4 weeks; q6w—every 6 weeks; PM—pegcetacoplan monthly; PEOM—pegcetacoplan every other month; RPE—incomplete retinal pigment epithelium; iRORA—incomplete outer retina atrophy; PRD—photoreceptors degeneration; FAF—fundus autofluorescence imaging; GAMM—generalized additive mixed-effect model; HRF—hyperreflective foci; LPR—local progression rate; PR—photoreceptor; RPE—retinal pigment epithelium; SD—spectral domain; EZ—ellipsoidal zone; DLS—double layer sign.; ↓ decreased; ↑ increased.

**Table 3 jpm-14-00990-t003:** Demographical, baseline characteristics, and outcomes of the patients in the analyzed studies.

Study	Age (yrs, Mean ± SD, for Each Study Arm)	Sex F (*n*, %)	Race (White *n*, %)	CNV in Fellow Eye (*n*, %)	Initial BCVA (Letters, Mean ± SD)	LLD (Letters, Mean ± SD)	Initial GA Area (Mean ± SD, mm^2^)	Initial GA Area (Mean Square Root ± SD, mm)	Change in Square Root GA Area (Mean ± SD, mm)	Change in GA Area (mm^2^)	Change in BCVA (Letters)	Change in LLD (Letters)	Targeted Genetic Analysis
Yehoshua Z [25], 2014 (COMPLETE study)	79 ± 7; 81 ± 6;	No info	No info	No info	71.3 ± 7.8; 78.6 ± 5.2;	No info	7.3 ± 4.8; 4.6 ± 3.6	2.5 ± 0.9; 2.02 ± 0.74	0.37 ± 0.21; 0.37 ± 0.22	No info	0.7 ± 7.2; 2.9 ± 7.0	No info	No correlation with tested alleles
Yaspan BL [27], 2017 (MAHALO Study)	80.4 ± 7.2; 77.1 ± 7.3; 78.5 ± 7.39	28 (66.7); 18 (43.9); 24 (60.0)	40 (95.2); 41 (100.0); 40 (100.0)	No info	47.6 ± 12.8; 49.5 ± 11; 45.9 ± 13.4	No info	8.5 ± 3.8; 8.5 ± 4.9; 8.8 ± 4.1	3.3 ± 1.5; 3.3 ± 1.9; 3.4 ± 1.6	2.3 ± 0.1 3.1 ± 0.1 2.9 ± 0.08	2.2 ± 1.3; 3 ± 1.9; 2.8 ± 2	−3.3 ± 1.9; −1.4 ± 1.9; −4.9 ± 1.9;	No info	CFH, C2/CFB- no correlation; carriers of the CFI risk allele associated with GA progression and response to IVL
Holz FG [28], 2018 (SPECTRI and CHROMA studies)	78.0 ± 8.0; 77.4 ± 7.9; 78.5 ± 8.3	379 (60.4); 375 (59.8); 377 (60.2)	608 (97.1); 608 (96.8); 611 (97.4)	No info	66.1 ± 9.8; 66.0 ± 9.9; 66.0 ± 9.9	30.1 ± 15.7; 29.7 ± 16; 29.6 ± 16.3	8 ± 4 8.1 ± 3.9 8.3 ±4.2	No info	0.3 ± 0.0 0.3 ± 0.0 0.3 ± 0.0	1.9 ± 0.0 2.05 ±0.0 2.05 ± 0.0	−4.1 ± 0.5 −4.9 ± 0.5 −4.9 ± 0.5	No info	CFI, CFH, C2/CFB not related to GA progression and response to IVL
Liao DS [29], 2020 (FILLY study)	79.6 ± 7.5; 80.9 ± 7.5; 78.4 ± 7.4	55 (64.0); 50 (63.3); 49 (60.5)	84 (97.7); 76 (96.2); 81(100.0)	36(41.9); 28(35.4); 29(35.8)	59.8 ± 15.7; 58.4 ± 16.0; 59.8 ± 17.2	23.5 ± 14.5; 27.1 ± 15.7; 26.2 ± 17.1	8 ± 3.8; 9 ± 4.47; 8.2 ± 4	2.7 ± 0.6; 2.9 ± 0.7; 2.8 ± 0.7	0.2 ± 0.02; 0.2 ± 0.02; 0.3 ± 0.02	2.35 ± 0.02; 2.6 ± 0.02; 3.05 ± 0.02	−7.7 ± 0.5; −8.8 ± 0.6; −6.4 ± 0.5	−2.7 * −4.6 −4.2	rs2230199 in C3 and rs3750846 in ARMS2 correlated with GA growth independent of the treatment arm
Heier JS [38], 2023 (OAKS and DERBY Studies)	OAKS: 78.8 ± 7.2 78.1 ± 7.7 7 8.6 ± 7.3 DERBY: 78.7 ± 6.9 79.2 ± 7.1 78.6 ± 7.3	OAKS: 125 (62%) 117 (57%) 133 (64%) DERBY: 118 (59%) 120 (60%) 123 (63%)	OAKS: 185 (92%) 189 (92%) 188 (91%) DERBY: 187 (93%) 186 (93%) 188 (96%)	No info	OAKS 61 ± 15.3 58.2 ± 17 57.6 ± 16.6 DERBY: 59.5 ± 17.4 58.7 ± 16.1 59 ± 16.9	OAKS: 26.7 ± 16.8 25.7 ± 17.6 24.9 ± 17.4 DERBY: 27.3 ± 17.7 25.6 ± 16.4 25.7 ± 16.5	OAKS: 8.1 ± 3.9 8.3 ± 3.9 8.2 ± 3.7 DERBY: 8.3 ± 4.1 8.2 ± 3.9 8.2 ± 4.2	No info	OAKS: 1.56 ± 0.08 1.65 ± 0.08 1.97 ± 0.08 DERBY: 1.73 ± 0.08 1.76 ± 0.07 1.96 + 0.1	OAKS: 3.1 ± 0.1 3.3 ± 0.1 4·0 ± 0.15 DERBY: 3.2 ± 0.1 3.3 ± 0.1 3.9 ± 0.2	−7.9 ± 0.7 −8.8 ± 0.7 −6.9 ± 0.7	No info	Not assessed
Pattel SS [40], 2021 (GATHER 1 study)	78.8 ± 10.2 78.2 ± 8.8 79.2 ± 8.3 78.2 ± 9.0	45 (67.2) 79 (71.8) 58 (69.9) 61 (72.6)	67 (100) 107 (97.3) 82 (98.8) 82 (97.6)	No info	70.2 ± 10 69 ± 10.4 69.5 ± 9.8 68.3 ± 11.0	33.5 34.5 32.7 34.4	7.3 ± 3.8 7.4 ± 3.8 7.9 ± 4.2 7.4 ± 3.9	2.6 ± 0.7 2.6 ± 0.7 2.7 ± 0.7 2.6 ± 0.7	0.3 ± 0.07 0.4 ± 0.07 0.3 ± 0.07 0.4 ± 0.07	No info	−7.9 ± 2.6 −9.3 ± 2.6 −3.8 ± 3.1 −3.5 ± 3	No info	Not assessed
Khanani MA [41], 2023 (GATHER 2 study)	77; 77	154 (68%) (treated arm) 156 (70%, sham)	182 (81%) 186 (84%)	No info	70.9 ± 8.9 71.6 ± 9.4	29.1 ± 19.7 32.6 ± 19.6	7.5 ± 4 7.8 ± 3.9	2.6 ± 0.7 2.7 ± 0.7	1.6 ± 0.1 2.3 ± 0.1	1.7 ± 0·2 2.1 ± 0·2	1.3 ± 1·4 0.9 ± 1·5	−3.0 ± 2.1 −1.7 ± 1.4	Not assessed

**Table 4 jpm-14-00990-t004:** Comparison between treated arms and sham groups, based on pooled data in the reviewed studies.

	Complement Inhibitor Treated Arm (*n* = 2684)	Sham Arm (*n* = 1476)	*p*-Value *
Age (years)	78.1 ± 7.8	77.8 ± 7.9	0.67
White (*n*, %) **	2642 (96.8%)	1376 (91.4%)	0.23
Female **	1524 (56.7%)	942 (63.8%)	0.38
GA area (mm^2^)	8.0 ± 3.9	8.1 ± 4	0.69
Square root GA area (mm)	2.7 ± 0.9	2.7 ± 0.8	1.00
NL-BCVA	63.7 ± 13.4	64.2 ± 13.7	0.24
LLD	28.6 ± 16.6	29.1 ± 17.1	0.35
Change in GA area (mm^2^)	2.4 ± 0.7	2.7 ± 0.8	<0.0001
Change in square root GA area (mm)	0.29 ± 0.05	0.32 ± 0.06	<0.0001
Change in NL-BCVA	−5.07 ± 3.03	−4.9 ± 3.1	0.08

Note: * by Chi-square test; ** based on available info.

**Table 5 jpm-14-00990-t005:** Effects of complement inhibitors vs. sham on BCVA: meta-analysis for the continuous measure.

Study	N1 (Treated)	BCVA Change Treated Group, Mean (SD)	N2 (Sham)	BCVA Change Sham Group, Mean (SD)	SMD	SE	95% CI	*p*	Weight (%)	
									Fixed	Random
MAHALO [27]	42	−4.1 (0.5)	40	−4.9 (0.5)	1.585	0.251	1.085 to 2.085		4.23	14.22
SPECTRI CHROMA [28]	628	−7.9 (2.6)	626	0.9 (1.5)	−4.071	0.0990	−4.265 to −3.877		27.30	14.32
FILLY [29]	84	−7.7 (0.5)	80	−6.4 (0.5)	−2.588	0.211	−3.005 to −2.171		6.00	14.26
OAKS [38]	202	−7.9 (0.7)	207	−6.9 (0.7)	−1.426	0.111	−1.643 to −1.209		21.86	14.32
DERBY [38]	201	−7.7 (0.8)	195	−6.8 (0.7)	−1.194	0.109	−1.408 to −0.980		22.54	14.32
GATHER 1 [40]	67	−3.3 (1.9)	134	−9.3 (2.6)	2.508	0.195	2.124 to 2.892		7.06	14.27
GATHER 2 [41]	225	1.3 (1.4)	223	−4.9 (1.9)	3.711	0.156	3.405 to 4.018		11.02	14.29
Total (fixed effects)	1449		1505		−1.194	0.0517	−1.296 to −1.093	<0.001	100.00	100.00
Total (random effects)	1449		1505		−0.213	1.061	−2.294 to 1.867	0.841	100.00	100.00

Note: Test for heterogeneity: Q: 2368.6; DF: 6; *p* < 0.0001; I^2^: 99.7%.

**Table 6 jpm-14-00990-t006:** Effects of intravitreal complement inhibitors vs. sham on GA area change: meta-analysis for continuous outcomes.

Study	N1 (Treated)	N2 (Sham)	Root Square GA Change: Treated vs. Sham	SMD	SE	95% CI	t	*p*	Weight (%)	
									Fixed	Random
MAHALO [27]	42	40	2.3 (0.08) vs. 2.9 (0.08)	−7.429	0.620	−8.663 to −6.195			0.49	13.19
SPECTRI CHROMA [28]	628	626	0.3 (0.0) vs. 0.3 (0.0)	0.000	0.0564	−0.111 to 0.111			59.19	14.59
FILLY [29]	84	80	0.2 (0.02) vs 0.3 (0.02)	−4.977	0.316	−5.600 to −4.353			1.89	14.21
OAKS [38]	202	207	1.56 (0.08) vs. 1.97 (0.08)	−5.116	0.204	−5.517 to −4.714			4.52	14.44
DERBY [38]	201	195	1.73 (0.08) vs. 1.97 (0.1)	−2.650	0.138	−2.920 to −2.379			9.96	14.53
GATHER 1 [40]	67	134	0.3 (0.07) vs. 0.4 (0.07)	−1.423	0.165	−1.749 to −1.098			6.92	14.49
GATHER 2 [41]	225	223	1.6(0.7) vs. 2.3(0.1)	−1.395	0.105	−1.601 to −1.188			17.03	14.56
Total (fixed effects)	1449	1505		−0.962	0.0434	−1.047 to −0.877	−22.146	<0.001	100.00	100.00
Total (random effects)	1449	1505	2954	−3.219	0.722	−4.636 to −1.803	−4.457	<0.001	100.00	100.00

**Table 7 jpm-14-00990-t007:** Systemic and ocular adverse events following complement inhibitors therapy.

Study, Year	Systemic SAE (*n*,%)	Ocular SAE (*n*,%)	IOP > 30 mmHg (*n*,%)	Endophthalmitis (*n*, %)	Non-Infectious Ocular Inflammation	De Novo CNV in the Study Eye
Yaspan BL [27], 2017	11 (25.6) 10 (22.7) 15 (35.7)	0 3 (6.8) 1 (2.4)	Not reported	None	Not reported	Not reported
Holz FG [28], 2018	120 (19); 84 (13.9); 103 (16.6)	39 (6.2); 38 (6.1); 17 (2.7)	52 (8.3); 35 (5.6); 2 (0.3%)	5 (0.04% per no. of injections) 5 (0.4% per no. of patients)	Not reported	Study eye: 7 (1.1); 12 (1.9); 12 (1.9); Fellow eye: 8 (1.3); 10 (1.9); 11 (1.9)
Liao DS [29], 2020	19 (22.1);	4 (4.7); 2 (2.5); 1 (1.2)	1 (1.2); 1 (1.3); 0	2 (2.3); 1 (1.3); 0	Not reported	18 (20.9); 7 (8.9); 1 (1.2)
Heier JS [38], 2023	No info	11 (2.6) 8 (1.9) 3 (0.7)	No info	OAKS: 2 (1); 2 (1); 0; DERBY: 0; 0; 0	OAKS: 11 (5); 3 (1); 1 (0) DERBY: 5 (2); 6 (3); 0	OAKS: 24 (11); 16 (8); 4 (2) DERBY: 27 (13); 12 (6); 9 (4)
Pattel SS [40], 2023	No info	1 (1.5) 1 (1.2) 0	No info	0; 0; 0	1 (1.5) in 2 mg arm	No info
Khanani AM [41], 2023	29 (13) 35 (16)	5 (2) 1 (<1)	No info	0; 0	No info	2 (1) 1 (0.5)

SAE—serious adverse effect; IOP—intraocular pressure; CNV—choroidal neovascularization.

## Data Availability

No new data were created.

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
