# Peer review of "Complement Inhibitors for Geographic Atrophy in Age-Related Macular Degeneration—A Systematic Review"

_jpm, 2024, doi:10.3390/jpm14090990_

Round 1

Reviewer 1 Report

Comments and Suggestions for Authors

This is good systematic review and meta-analysis dealing with the use of complement inhibitors in AMD.

I recommend performing the following:

1- In the first paragraph of the results: Please mention that the review was carried out for papers published in the last decade: from 2014-2024.

2- In Table 2: PLease ensure consistency in organizing the studies according to the year of the study.

3- Please mention the software and its origin that the meta-analysis was performed with and the Forest Plot was generated from.

4- The heterogeneity parameters such as the Q-value and I2 are v. high. PLease interpret the reasons for this heterogenity.

Comments on the Quality of English Language

Acceptable

Author Response

This is good systematic review and meta-analysis dealing with the use of complement inhibitors in AMD.

I recommend performing the following:

  • In the first paragraph of the results: Please mention that the review was carried out for papers published in the last decade: from 2014-2024.

R: We have added this correction, as recommended.

  • In Table 2: PLease ensure consistency in organizing the studies according to the year of the study.

R: Dear reviewer, we have reordered the studies by the year of publication in all tables, and updated the references accordingly.

  • Please mention the software and its origin that the meta-analysis was performed with and the Forest Plot was generated from.

R: The meta-analysis was performed by MedCalc Software© (2024 MedCalc Software Ltd, Ostend, Belgium). We have added this information in the text.

4- The heterogeneity parameters such as the Q-value and I2 are v. high. PLease interpret the reasons for this heterogenity.

R: Thank you for the suggestion, we have added a paragraph to discuss this aspect, the possible causes and the impact upon the interpretation of the results.

Reviewer 2 Report

Comments and Suggestions for Authors

interesting subject, english is ok

The thing that disturbs me is that this article really looks like the one published by Abidi et al in Ophthalmol Sci  2022 :"aclinical and preclinical assessment of clinical trials for age related macular degeneration, this is why I considered originalty low.

And these authors are not cited in references

Author Response

interesting subject, english is ok

The thing that disturbs me is that this article really looks like the one published by Abidi et al in Ophthalmol Sci  2022 :"aclinical and preclinical assessment of clinical trials for age related macular degeneration, this is why I considered originalty low.

And these authors are not cited in references

R: Dear reviewer, thank you for your recommendation. We have added this valuable paper in the reference list.

We have included in our review studies published after 2022, and perform a metanalysis to compare the results, in terms of functional and anatomic outcomes. In the light of the recent approval by FDA of the 2 complement inhibitors for age related macular degeneration with geographic atrophy, we do hope you will consider our work an useful up-to-date on this subject.

Reviewer 3 Report

Comments and Suggestions for Authors

The manuscript «Complement Inhibitors for Geographic Atrophy in Age-Related Macular Degeneration – systematic review» by AM Dascalu and colleagues provides a non-comparative summary of existing RCTs and a random effects model to predict the treatment outcomes for the treatment of GA. The manuscript is methodologically sound, but not all readers will be confident with the interpretation of outcomes of random effect models, so that the interpretation of fixed and random effects should be added to methods. Moreover, the outcomes represented in tables 5 and 6, as well as figures 2 and 4 deserve to be elaborated including the addition of the parameter to which the standard mean differences refer in both figures (change in BCVA and change in GA area).

-       In my opinion, the Complete study may be reported, but should be excluded from the analysis, given the imbalance in baseline BCVA and lesion size which largely explain the absence of a therapeutic effect of Eculizumab (lines 254-6). This study is so different in baseline characteristics, that comparability to the other 8 studies is not obviously given the numerically the NOS score is sound (see table 1).

-       The manuscript does not report the differences in inclusion criteria and their bearing onto the outcomes of these studies, i.e. lesion size and location, exclusion of nAMD, as well as BCVA window which is fundamental for understanding the outcomes (lines 325 ff).

-       As evidenced by the variability of change in lesion size and the absence of functional advantages, other effect markers, such as PR and RPE loss, EZ impairment, progression from iRORA to cRORA, may be more important to individualize treatment and to predict the long-term functional outcomes (lines 419-21). Are these possible new biomarkers that might be more useful for defining and controlling the treatment success (lines 421-5).

-       Personally, I would highlight interesting findings in tables 2 and 3 by bold letter format.

-       Since, as mentioned by the authors in the discussion, monthly dosing may lead to optimal outcomes, but will not be realistic in terms of patient adherence, the reader deserves to learn about the outcomes of lower and every other month dosing. This information should be added to lines 265-8.

-       Are ERP EPR and RPE the same? Pls unify and explain abbreviations.

-       The sentence in line 292 should be supplemented by a reference.

-       Safety is crucial given the need for a long-term treatment for a chronic disease which progresses less rapidly under treatment. The two clinically most important safety markers should be discussed, which are switch to neovascular disease and intraocular inflammation. The risk to develop nAMD is increased by 3-9x for Peg, and 3-4x for Ava over 2 years. What does that imply for long-term treatment? Please discuss how to escape this risk and how to handle these complications. Should patients with nAMD also be offered complement inhibitors for their underlying degenerative disease?

-       What is the message of lines 396-7, what the clinical impact of C3 depositions at the level of which target cells?

-       Some abbreviations are lacking, for example ERP, EPR or MAC, a lot of spelling errors need to be corrected.

In conclusion, as the authors state in the last sentences of this manuscript, case selection or predicting the treatment response on an individual level are crucial to justify the individual and society burden associated with this disease and its treatment. The discussion has not covered this clinically important points, while complement inhibition as a therapeutic option for nAMD (chapter 4.2.1) should also be discussed with reference to a 4x increase the risk of switch to nAMD under intravitreal complement inhibitors.

Taken together, an interesting overview on available RCTs to the topic (see tables), but no support in interpreting the outcomes and namely no compariosn of different studies, i.e. Pegcetacoplan and Avacincaptad, both of which have recently been approved by the FDA, but for well-considered reasons not by the European authorities. Would recommend to completely rewrite the discussion accordingly. 

Comments on the Quality of English Language

Please have a language and spelling check run throughout. Examples justifying thie request are found exemplarily in lines 55, 61, 85 ... 

Author Response

The manuscript «Complement Inhibitors for Geographic Atrophy in Age-Related Macular Degeneration – systematic review» by AM Dascalu and colleagues provides a non-comparative summary of existing RCTs and a random effects model to predict the treatment outcomes for the treatment of GA. The manuscript is methodologically sound, but not all readers will be confident with the interpretation of outcomes of random effect models, so that the interpretation of fixed and random effects should be added to methods. Moreover, the outcomes represented in tables 5 and 6, as well as figures 2 and 4 deserve to be elaborated including the addition of the parameter to which the standard mean differences refer in both figures (change in BCVA and change in GA area).

Response: Dear reviewer, many thanks for all your time spent reviewing our manuscript, and the helpful suggestions and comments. We have carefully revised our manuscript, according to your indications.

We have added the required info in Table 5 and 6. We have added a paragraph to discuss the possible causes for the high heterogeneity observed, and the impact upon the interpretation of the results. We removed the COMPLETE study from the analysis, as recommended.

We also agree with your suggestion, to focus more on the 2 therapeutic agents which recently were approved by FDA, Pegcetacoplan and Avancicaptad. For this reason, we calculated the combined means and SD for the change in GA observed in the subgroups studies, and compared the pooled results.

-       In my opinion, the Complete study may be reported, but should be excluded from the analysis, given the imbalance in baseline BCVA and lesion size which largely explain the absence of a therapeutic effect of Eculizumab (lines 254-6). This study is so different in baseline characteristics, that comparability to the other 8 studies is not obviously given the numerically the NOS score is sound (see table 1).

Response: Thank you for this observation! We have added this comment and removed the COMPLETE study from the quantitative analysis, as recommended.

-       The manuscript does not report the differences in inclusion criteria and their bearing onto the outcomes of these studies, i.e. lesion size and location, exclusion of nAMD, as well as BCVA window which is fundamental for understanding the outcomes (lines 325 ff).

Response: Dear reviewer, thank you for the useful comment! We have added a paragraph to discuss the differences in terms of GA size and location, BCVA and the presence of neovascular AMD in the fellow eye. We totally agree these factors may cause the differences in the observed outcomes.

-       As evidenced by the variability of change in lesion size and the absence of functional advantages, other effect markers, such as PR and RPE loss, EZ impairment, progression from iRORA to cRORA, may be more important to individualize treatment and to predict the long-term functional outcomes (lines 419-21). Are these possible new biomarkers that might be more useful for defining and controlling the treatment success (lines 421-5).

Response: We totally agree, we emphased this idea in the discussions.

-       Personally, I would highlight interesting findings in tables 2 and 3 by bold letter format.

Response: Thank you for the suggestion, we did so.

-       Since, as mentioned by the authors in the discussion, monthly dosing may lead to optimal outcomes, but will not be realistic in terms of patient adherence, the reader deserves to learn about the outcomes of lower and every other month dosing. This information should be added to lines 265-8.

Response: We added a paragraph to comparatively discuss the comparative effects of monthly dosing versus every other month therapy, for each approved agent (pegcetacoplan and avacincaptad pegol), as reuired, in Section 3.3. Interestingly, in longer term (24 months follow up) the differences become less significant, and this aspect may be useful to know in clinical practice.

As well, we added this idea in the Discussions.

-       Are ERP EPR and RPE the same? Pls unify and explain abbreviations.

Response: Yes, we have corrected.

-       The sentence in line 292 should be supplemented by a reference.

Response: We added the missing reference.

-       Safety is crucial given the need for a long-term treatment for a chronic disease which progresses less rapidly under treatment. The two clinically most important safety markers should be discussed, which are switch to neovascular disease and intraocular inflammation. The risk to develop nAMD is increased by 3-9x for Peg, and 3-4x for Ava over 2 years. What does that imply for long-term treatment? Please discuss how to escape this risk and how to handle these complications. Should patients with nAMD also be offered complement inhibitors for their underlying degenerative disease?

Response: We have added a paragraph in the Discussion section to discuss this idea, as suggested.

-       What is the message of lines 396-7, what the clinical impact of C3 depositions at the level of which target cells?

Response: We added a detailed explanation: inhibiting complement at the C5 level may preserve some functions related to C3 fragments, that, on one hand may provide some anti-infectious protection (preventing ocular infections), but, on the other hand, may still associate some pro-inflammatory activity at the retinal level.

-       Some abbreviations are lacking, for example ERP, EPR or MAC, a lot of spelling errors need to be corrected.

 Response: We have carefully re-checked the manuscript, added the missing explanations for abbreviations and corrected the spelling errors.

In conclusion, as the authors state in the last sentences of this manuscript, case selection or predicting the treatment response on an individual level are crucial to justify the individual and society burden associated with this disease and its treatment. The discussion has not covered this clinically important points, while complement inhibition as a therapeutic option for nAMD (chapter 4.2.1) should also be discussed with reference to a 4x increase the risk of switch to nAMD under intravitreal complement inhibitors.

Taken together, an interesting overview on available RCTs to the topic (see tables), but no support in interpreting the outcomes and namely no compariosn of different studies, i.e. Pegcetacoplan and Avacincaptad, both of which have recently been approved by the FDA, but for well-considered reasons not by the European authorities. Would recommend to completely rewrite the discussion accordingly. 

Response: Dear reviewer, we have revised the results and discussions sections, carefully following your indications. Thank you for your kind contribution to increase the value our manuscript!

Comments on the Quality of English Language

Please have a language and spelling check run throughout. Examples justifying thie request are found exemplarily in lines 55, 61, 85 ... 

Response: We have carefully revised English spelling and grammar in the entire manuscript.

Round 2

Reviewer 3 Report

Comments and Suggestions for Authors

Happy with the changes

Comments on the Quality of English Language

 the linguistic quality namely of new parts may benefit from quality checking

Author Response

Dear reviewer,

We are happy that all issue related to the scientific part of the manuscript were solved.

At your recommendation, we have submitted our manuscript to the English editing service provided by MDPI, to improve English in our paper.

Here we upload the revised manuscript.